# Personality Determinants of Eating Behaviours among an Elite Group of Polish Athletes Training in Team Sports

**DOI:** 10.3390/nu15010039

**Published:** 2022-12-21

**Authors:** Maria Gacek, Agnieszka Wojtowicz, Adam Popek

**Affiliations:** 1Department of Sports Medicine and Human Nutrition, Faculty of Biomedical Sciences, University of Physical Education in Kraków, 31-571 Kraków, Poland; 2Department of Psychology, Faculty of Social Sciences, University of Physical Education in Kraków, 31-571 Kraków, Poland; 3Bronisław Markiewicz State Higher School of Technology and Economics in Jarosław, 37-500 Jarosław, Poland

**Keywords:** athletes’ diet, the big five model, nutritional behaviours, personality traits, team sports disciplines

## Abstract

The nutritional behaviours of athletes are determined by environmental and individual factors. The aim of the research was to analyse the personality determinants of the eating behaviour among an elite group of Polish athletes training in team sports. The research was conducted among 213 athletes, using a proprietary validated nutritional behaviour questionnaire and the Neuroticism Extraversion Openness-Personality Inventory-Revised (NEO-PI-R Personality Inventory). Statistical analysis was performed with the use of Pearson’s linear and Spearman’s signed rank correlation coefficients, as well as multiple regression evaluation, assuming the significance level of α = 0.05. It was shown that the overall index of proper eating behaviour increased with increasing neuroticism (r = 0.132) and decreased with increasing openness to experience (r = −0.143). When assessing individual nutritional behaviours, it was indicated, among others, that with increasing neuroticism, athletes more often avoided energy drinks (R = 0.173), and with increasing extraversion, they more frequently consumed vegetables at least twice a day (R = 0.154). At the same time, the consumption of raw vegetables (R = −0.153), 2–3 portions of vegetables and fruit per day (R = −0.157) and the limitation of sweet and salty snacks (R = −0.152) decreased along with an increase in openness. On the other hand, with increasing conscientiousness, the regular consumption of meals (R = 0.186), dairy products (R = 0.143) and the reduction of sweet and salty snacks (R = 0.148) increased. The model built on the basis of the Big Five personality traits explained, to a very large extent (approx. 99%), variance concerning the general index of normal eating behaviour among athletes. The predictive significance of the personality traits presented in the Big Five model was demonstrated in relation to the quality of nutritional choices among Polish athletes competitively training in team sports, which may be used to personalise the dietary education of athletes.

## 1. Introduction

A key factor determining health and exercise capacity as well as effective post-exercise recovery is the implementation of a varied and balanced diet, taking products with high nutritional density into account. Athletes demonstrate an increased demand for energy, selected nutrients (carbohydrates, proteins, B vitamins, antioxidants and mineral salts) and fluids [1,2]. One of the proposals of sports nutrition is the Swiss pyramid for individuals undertaking increased physical activity, which begins with water and other unsweetened beverages, and ends in sweets, salty snacks and sweetened drinks. Intermediate levels of the pyramid are occupied by: vegetables and fruits, whole grain cereals, legumes, other protein products and vegetable fats, recommended for consumption in various amounts and at a specific frequency [3]. The mentioned model of nutrition emphasises the special significance of water and other unsweetened beverages for the regulation of water and electrolyte balance as well as vegetables and fruit (products with low and medium glycaemic index, rich in dietary fibre, potassium, magnesium, B vitamins and antioxidants) for maintaining antioxidant status and acid–base balance under conditions of vigorous physical exercise [3,4,5,6]. The diet of athletes can be supplemented with measures of scientifically proven effectiveness and safety of use, including special food for athletes (including isotonic drinks and energy gels), medical supplements (i.e., multivitamins, vitamin D, calcium, iron, zinc, probiotics) and ergogenic aids improving physical performance (inter alia, caffeine and creatine), classified in group A according to the opinion of experts from the Australian Institute of Sport and IOC [7,8,9].

The nutritional behaviour of athletes Is dynamic and determined by numerous factors, including environmental and individual [10,11]. Important nutritional behaviour determinants are psychological factors, including personality traits. Personality is treated as an internal regulatory system that allows adaptation to selected situations and the environment as well as internal integration of thoughts, feelings and behaviours [12]. One of the models regarding the conception and description of personality is the Five-Factor model of personality by Costa and McCrae which, in the last few decades, has become one of the dominant paradigms in trait psychology [13]. The so-called Big Five model by Costa and McCrae [14] consists of five main dimensions of personality, including: neuroticism, extraversion, openness to experience, agreeableness and conscientiousness. The set of Big Five dimensions creates the possibility of a multifaceted characteristic and interpretation of personality in terms of five main domains: general activity and contact, emotional balance, attitude towards people, attitude towards tasks and general reference to the world and new experiences [12,13,14].

Research on the relationships between personality traits of the Big Five model and diet has been carried out on various continents and in numerous countries, including New Zealand among young adults, in West Africa among students of the University of Ghana and in Europe among Polish and Spanish physical education students [15,16,17]. In Poland, research on the psychological determinants of eating behaviours and the use of ergogenic aids by athletes has mainly concerned personal resources, including the general sense of self-efficacy and dispositional optimism [18,19,20,21,22].

Due to the importance of diet quality for the health and exercise capacity of athletes, the complexity of nutritional behaviour determinants, the lack of exploration of the subject and the ambiguity of the results achieved in previous studies, research on personality determinants of athletes’ diet has been undertaken. The aim of the research was to analyse the personality determinants of eating behaviours among an elite group of Polish athletes training in team sports. The correlations between the personality traits of the Big Five model and the implementation of the qualitative recommendations of the Swiss sports nutrition pyramid were assessed. The following research questions were posed: (1) What are the nutritional behaviours of athletes? (2) What are the personality traits of athletes? (3) What are the relationships between the personality traits and eating behaviours of the athletes? A research hypothesis was adopted stating that personality traits are related to nutritional behaviours, while correct nutritional choices are favoured by high conscientiousness and low neuroticism.

## 2. Materials and Methods

### 2.1. Participants

The research was carried out among a group of 213 Polish athletes (males) professionally training in team sports, including basketball (*n* = 54), volleyball (*n* = 53), football (*n* = 53) and handball (*n* = 53). The basic criterion for selecting people into the study group was practicing sports at a competitive level, at the level of the highest league in Poland, for at least 3 years. The basic criterion for exclusion was belonging to the lower league class and failure to meet the criterion of minimum sports experience (3 years). The studied players, in relation to the current classification of the level of activity and sports abilities [23], can be assigned to Tier 3 (Highly Trained/National Level). The research was conducted at 50 sports clubs (18 basketball, 19 volleyball, 13 football and 10 handball). The study comprised athletes aged 18–38 (M = 26.1; SD = 4.5), with sports experience in the range of 3–20 years (M = 8.2; SD = 4.5). The research study was performed in accordance with the 1964 Declaration of Helsinki, after obtaining the participants’ written informed consent, while the research protocol was approved by the Bioethics Committee at the Regional Medical Chamber in Krakow (No. 105/KBL/OIL/2021).

### 2.2. Instruments

To assess the diet, a proprietary, validated questionnaire of nutritional behaviour was used, referring to the qualitative recommendations of the Swiss nutrition pyramid for individuals performing increased physical activity presented in literature on the subject [3]. The questionnaire consists of 23 statements (items) on rational eating behaviours, with a 5-point Likert scale of answers (from 1–5, from “definitely not”, “probably not”, “hard to say”, “rather yes” to “definitely yes”). The questions concerned: regularity of consuming at least 3 meals a day, the recommended frequency of consuming vegetables and fruits, whole grains, dairy products, other nutritional sources of protein, adequate hydration before, during and after training, preferred fats and limiting non-recommended products (sweets, fast foods, carbonated and non-carbonated sweetened beverages and energy drinks). Based on the results of the questionnaire, the degree of implementing individual nutritional recommendations and the overall index of rational eating behaviour were assessed (on a scale of 1–115 points, assuming that the higher the index, the more intense rational eating behaviour). The questionnaire was validated. The validity of the test was assessed via the re-test method (*n* = 32). The value of the linear correlation coefficient was calculated and the null hypothesis, H0: r = 0, was tested using the Student’s *t*-test, obtaining a result allowing us to confirm the reliability of the scale (r = 0.388; *p* = 0.036). The internal consistency of the scale was also good (Cronbach’s α coefficient totalled 0.79).

To assess personality traits from the Five-Factor model, the Neuroticism Extraversion Openness-Personality Inventory-Revised (NEO-PI-R Inventory) by P.T. Costa and R.R. McCrae was used [14,24]. The NEO-PI-R includes 240 statements, the truthfulness of which is assessed on a 5-point scale (from “completely disagree” to “completely agree”). The statements refer to 5 personality factors (scales), and within each of them, to 6 components (subscales). These include: neuroticism (anxiety, hostility/anger, depression, self-consciousness, impulsiveness/immoderation, vulnerability to stress/fear/learned helplessness), extraversion (warmth/kindness, gregariousness/sociability, assertiveness, activity level/lively temperament, excitement seeking and positive emotion), openness to experience (fantasy/imagination, aesthetics/artistic interest, feelings/emotionality, action/adventurousness/exploration, ideas/intellectual interest/curiosity, values/psychological liberalism/tolerance to ambiguity), agreeableness (trust in others, straightforwardness/morality, altruism, compliance/cooperation, modesty, tendermindedness/sympathy) and conscientiousness (competence/self-efficacy, order(liness)/organising, dutifulness/sense of duty/obligation, achievement striving, self-discipline/willpower, deliberation/consciousness). The reliability of measurements regarding the 5 dimensions in the Polish adaptation of the questionnaire is adequate and amounts to: neuroticism −0.86, extraversion −0.85, openness to experience −0.86, agreeableness −0.81 and conscientiousness −0.85 [24]. Due to the fact that the relationships between the characteristics (subscales) of the basic dimensions of personality with nutritional behaviours did not reach a level of statistical significance, the paper presents only the results on the basic 5 dimensions of personality from the Big Five model.

### 2.3. Statistical Analysis

The collected numerical material was statistically analysed using the Statistica 13.3 package. Basic statistical measures (M—arithmetic mean, Me—median, SD—standard deviation, Q25—lower quartile Q75—upper quartile, minimum and maximum) were calculated. Statistical analysis was performed using Pearson’s linear and Spearman’s signed rank correlation coefficients (depending on the nature of the variables). Multiple regression analysis was also performed to see which of the variables could explain the level concerning the overall index of normal nutrition behaviours among athletes. In the calculations, a progressive stepwise regression procedure without constant terms was used. The analysis also included calculation of values for the multivariate determination coefficient (R^2^) and the standard error of estimate (s_y_), as well as the values of standardised partial regression coefficients, b *, which are a measure of the relative significance regarding particular features (X variables) in the model. The analyses were performed with the significance level of α = 0.05.

## 3. Results

### 3.1. Athletes’ Nutritional Behaviours

In Table 1, the implementation of nutritional recommendations is shown according to the suggestions of the Swiss pyramid for athletes. In the summary of affirmative (“rather yes” and “definitely yes”) as well as negative responses (“rather no” and “definitely not”), the surveyed athletes declared consuming at least three meals a day, adequate hydration during and after exercise and avoiding fast food products in the highest percentage (above 90%). A high percentage (over 80%) declared that they avoided carbonated and non-carbonated sweetened beverages as well as energy drinks in their diet. Approximately 50% declared regular consumption of meals and eating the most caloric meal before or after main training. Approx. 1/3 of the respondents ate fish 1–2 times a week, 1–2 portions of fruit daily, whole grain products at least twice a day, followed a varied diet and reduced animal fats in their diet. The athletes responded least positively to the questions on the consumption of raw vegetables at least once a day, and of milk and dairy products at least twice a day (Table 1).

Assessment of nutritional behaviours (according to the median value) allowed us to confirm that the athletes most closely followed the recommendations regarding the consumption of at least 3 meals a day, adequate hydration during exercise and preferring water for rehydration (Me = 5.00, “definitely yes”). The high level of implementation also concerned the regularity of meals and the consumption of the most caloric meal before or after the main training session (Me = 4.00, “rather yes”). Other behaviours were realized to a lesser extent (Me = 3.00, “hard to say”, i.e., neither yes nor no). The overall index of compliance with the recommendations of the Swiss pyramid in the studied group of athletes was 81.9 ± 4.64 points (out of a possible 115) (Table 2).

### 3.2. Athletes’ Personality Traits 

Among the personality dimensions of the Five-Factor model, the tested athletes obtained the highest results in terms of conscientiousness (M = 128.50), agreeableness (M = 123.20) and extraversion (M = 121.80), lower results regarding openness to experience (M = 115.00) and the lowest in terms of neuroticism (M = 72.15) (Table 3).

### 3.3. Relationships of Personality Traits with the Nutritional Behaviours of Athletes

It was shown that the overall index of proper eating behaviours increased with increasing neuroticism (r = 0.132) and decreased with increasing openness to experience (r = −0.143). As part of the assessment of individual eating behaviours, it was demonstrated that along with an increase in neuroticism, regular consumption of meals (R = 0.143), daily consumption of 2–3 portions of vegetables and fruit (R = 0.147) and avoidance of energy drinks increased (R = 0.173). At the same time, with increasing extraversion, the consumption of vegetables in at least two meals a day increased (R = 0.154). With the increase in openness, the consumption of raw vegetables at least once a day (R = −0.153), 2–3 portions of vegetables and fruit per day (R = −0.157) and limiting sweet and salty snacks (R = −0.152) experienced a decrease. As agreeableness rose, the consumption of cereal products also increased (R = 0.149), while adequate hydration decreased after the completion of training (R = −0.148). Moreover, as conscientiousness increased, so did the regular consumption of meals (R = 0.186), consumption of at least two servings of dairy products per day (R = 0.143) as well as limiting sweet and salty snacks (R = 0.148) (Table 4).

Multiple regression analysis (dependent variable: indicator of proper eating behaviours; predictors: personality traits of the Big Five model) showed that the full model consisting of all analysed personality traits explains 99.4% of the variance in the level of the rational nutrition behaviour index, with agreeableness (strongest correlation, b * = 0.467), extraversion, neuroticism and conscientiousness (proportional relationships) being significant predictors (Table 5).

## 4. Discussion

The discussed results of our research allowed us to demonstrate an average implementation level of qualitative nutritional recommendations for athletes, a varied intensity of personality dimensions (the Big Five model) and significant relationships between some personality dimensions and eating behaviours of Polish athletes competitively training in team sports.

Discussing nutritional behaviours, it is necessary to point to the average level of implementing the qualitative recommendations of the Swiss food pyramid among athletes (81.9 out of 115 points, i.e., 71%). At the same time, it should be highlighted that the level of implementation of individual nutritional recommendations is varied, including the highest level for consuming the recommended number of meals and adequate hydration in conditions of physical exertion. Proper fluid replenishment is a very important aspect of exercise physiology and sports nutrition related to the prevention of dehydration. In this regard, the significance of isotonic beverages in the effective hydration of the body, maintaining water and electrolyte balance and accelerating post-exercise glycogen resynthesis should be emphasised [1,2,9]. At the same time, however, among the surveyed athletes, a low level of implementing recommendations regarding the regularity of meals and the consumption of products with high nutritional density (including vegetables, especially raw, fruit, whole grain cereal products, dairy products and fish) and the reduction of animal fats and preferring vegetable fats in the diet was found. The noted nutritional mistakes could reduce nutritional value of the athletes’ diet. These increase the risk of deficiency concerning numerous nutrients, including polyphenols, antioxidant vitamins, B vitamins (including B1, B2, B6, B9) and vitamin D, some mineral salts (i.e., Mg, Ca), dietary fibre and omega-3 PUFAs. Such deficiencies may increase health risks, including those related to oxidative stress and its effects [1,6,25,26,27,28]. In a situation of high exposure to oxidative stress in conditions of vigorous physical exercise, a diet rich in food antioxidants (including vegetables and fruits) is an important aspect of sports nutrition [25,29,30,31].

The qualitative nutritional mistakes shown in the discussed research have also been described by other authors in many groups of athletes performing at various levels of sport, including Polish athletes of individual and team sports disciplines [32], field hockey players [33], English rugby players [34], amateur footballers [35], Spanish team sports players [36] and academic athletes [37]. Diet imbalances related to the improper consumption of products has also been described in numerous studies, among team and individual sports players [38], footballers from Cyprus [39], Polish basketball players [20], junior and senior football players [40], Australian footballers representing various sports levels [41,42] as well as professional footballers from the Dutch league [43]. The results of the quantitative evaluation of diet among elite Polish basketball players allowed us to confirm the low supply of many nutrients, including dietary fibres and vitamins (B1, B9, C) and mineral salts (potassium, calcium, iodine) [20]. Among elite representatives of Spanish team sports, low blood levels of vitamin D3 have been described [44]. In this context, the importance of medical supplements recommended for athletes with vitamin and mineral deficiencies, including multivitamins, vitamin D and calcium, cannot be denied [7,9].

In our study, it was also indicated that the elite group of Polish athletes training in team sports, in terms of the personality profile, was characterised by higher intensity regarding the dimensions of conscientiousness, agreeableness and extraversion, and the lowest concerning the level of neuroticism. It should be noted, however, that the authors’ intention was only performing a limited analysis of the athletes’ personality traits, to an extent constituting a background for the assessment of psychological determinants of nutritional behaviours. Nonetheless, it may be concluded that in other papers on the personality of athletes, different trends were shown, while all these studies allow confirmation of a low level of neuroticism among competitive athletes, including those performing team sports [45,46], and also among Polish athletes [47], especially those at a championship level [48,49]. The tendency towards low levels of neuroticism among athletes is important for interpreting its predictive role in the quality of athletes’ eating behaviours.

In our research, statistically significant (but weak) relationships were also demonstrated between the personality traits of the Big Five model and the eating behaviours of athletes. In terms of the overall index of proper nutritional behaviours, predictive significance of neuroticism (positive) and openness to experience (negative) was found. The performed multiple regression analysis confirmed a very high share regarding the analysed personality traits of the Big Five model in explaining the variance of the Swiss pyramid implementation index (approx. 99%). Positive correlations between extraversion and the consumption of vegetables were shown at the level of individual eating behaviours. However, the greatest number of positive associations was noted in terms of conscientiousness, the higher intensity of which was conducive to increasing the regularity of eating meals, consuming dairy products and reducing contraindicated products (sweet and salty snacks). The correlations between agreeableness and nutritional behaviours were not unambiguous.

The obtained results confirm the difficulties in definite assessment and interpretation of the correlations between the personality and nutritional choices of athletes. Noteworthy are the positive (but weak) correlations between neuroticism and rational eating behaviours. Although this dimension of personality is defined by a tendency towards emotional instability, increasing the risk of wrong food choices, it should be noted that the level of neuroticism in the tested athletes was low, which may explain the described tendencies in this regard. Other authors [47] drew attention to the low level of neuroticism among athletes, limiting the risk of making nutritional mistakes. The negative correlations between the dimension of openness and rational eating behaviours also require some comment, while openness is related, inter alia, to cognitive abilities and imagination (but also to greater boldness in undertaking experiments) [50]. The dimension of extraversion is related, among others, to positive emotionality, fostered rational choices in the consumption of vegetables, which should be (apart from fruit) the basis of a rational and balanced diet. However, it should be pointed out that extroverts are more focused on taking action than on the task and they tend to make mistakes, which may cause difficulty in explaining the relationship between extraversion and the quality of food choices [51,52]. The dimension of agreeableness, which may limit involvement in performed activities, was ambiguously related to food choices. Conscientiousness, defined as the ability to control stimuli and focus on a specific goal, favoured rational nutritional choices of athletes. Other studies conducted in the general population also allowed confirmation of a relationship between personality traits (especially neuroticism, extraversion and conscientiousness) and food choices. In this regard, it was shown that conscientiousness promoted, above all, the consumption of fruit and the restriction of meat, confectionery and sweetened beverages, e.g., by promoting moderation and refraining from emotional eating, and by applying dietary restrictions (as recommended). Neuroticism, in turn, through the mechanism of emotional eating, promoted, inter alia, the consumption of sweets and confectionery products. On the other hand, extraversion, due to the high sociability of extroverted people, promoted the consumption of sweets and confectionery [53].

The relationships between personality traits and food choices of various population groups were also the subject of other studies at various research centres. Research in this area among academic youth undertaking increased physical activity (Polish and Spanish students of physical education) showed partially similar regularities. Among physical education students, at the level of personality relations with the consumption of products recommended and not recommended in the diet, relationships were shown between high extraversion and the consumption of fruit as well as vegetables (but also sweets and confectionery products), between high conscientiousness and the consumption of vegetables and high neuroticism with a low consumption of sea fish [17]. The importance of personality traits for the quality of food choices was also described among academic youth from New Zealand (here, a positive correlation between extraversion and fruit consumption was noted) [15] and among students from Ghana (here, inter alia, correlations between extraversion and interest in new food products, associations of agreeableness with irregular consumption of meals and conscientiousness with a variety of diets and limiting sugar consumption were found) [16]. Research in the Indonesian population has shown that conscientiousness is the only personality dimension that is clearly, positively related to rational food choices [54]. In other studies conducted in Indonesia, aimed at assessing the relationship of personality traits with body mass index (BMI), it was shown that people with excess body mass were less extroverted (i.e., more introverted) than people with normal body weight, which may indirectly indicate better food choices (and greater physical activity) among more extroverted people [55]. Research on the correlations between personality traits, eating behaviours and BMI was also undertaken in the Australian population, which confirmed, among others, that the nutrition pattern based on plant foods (vegetables, fruits, legumes) and fish was positively associated with openness, conscientiousness and emotional stability, and that BMI was negatively correlated with conscientiousness as well as emotional stability, and positively with agreeableness [56]. Among Polish athletes, a relationship was found between conscientiousness and a lower body mass of female volleyball players [57]. However, the significance of correlations between personality traits and indices of nutritional status was noted among tennis players [58].

In a different study, a relationship has also been noted between other psychological features and diet quality of athletes. Research in this area allowed confirmation of the positive predictive importance of personal resources (general sense of self-efficacy, dispositional optimism and life satisfaction) for diet quality of athletes, including Polish American football players [21], athletes of individual disciplines [18] and elite Polish basketball players [19,20].

It may be summarised that studies on personality determinants of nutritional behaviours among various population groups sometimes yield varied and ambiguous results. Further research carried out by interdisciplinary teams is needed to explain the mechanisms of the dependencies in question, as also pointed out by other authors [54]. Nonetheless, it should be emphasised that the nutritional choices of athletes are dynamic and also depend on direct environmental factors [11]. The nutrition-related mistakes demonstrated among athletes justify the need to monitor diet and nutritional education, considering the individual impact of promoting a healthy diet. Assessing the predictive significance of personality traits included in the Big Five model in relation to the sports nutrition model should contribute to the effective rationalisation of diet through the individualisation of potential educational and dietary interactions, the importance of which is also noted by other authors [59].

The limitations of the work are primarily related to the failure to include demographic and sports variables (age, professional experience, discipline), taking one selected area of nutrition (eating behaviour) into account and the self-report nature of the research tools used. These as well as other limitations should set directions for further work, the aim and subject of which should be comprehensive assessment of personality determinants regarding various areas of sports nutrition (including the use of dietary supplementation and so-called alternative dietary strategies), considering demographic and sports characteristics (including sports level and type of discipline).

## 5. Conclusions

Among athletes competitively performing team sports, an average level of the proper nutritional behaviour index was shown, with the players fulfilling recommendations regarding the number of meals and fluid replenishment in conditions of physical exertion to the greatest extent. Nutrition-based mistakes, on the other hand, concerned irregular consumption of meals and insufficient intake of products with high nutritional density (including raw vegetables, fruit, whole grain cereal products, dairy products and fish), which could have reduced the nutritional and health quality of the diet.Among the personality dimensions of the Big Five model, high-class athletes training in team sports are characterised by a low level of neuroticism, which is important for assessing the relationship between neuroticism and diet.Significant correlations have been found between the personality traits of the Big Five model and the implementation of qualitative nutritional recommendations for athletes, with the overall index of proper nutritional behaviours increasing with increasing neuroticism, and decreasing with increasing openness (however, the strength of the correlations was weak). No correlations were noted between the overall index of rational food choices and the features (subscales) of the basic personality dimensions concerning the Big Five model. At the level of individual eating behaviours, the most positive correlation was related to conscientiousness, which was conducive to more rational food choices.The obtained results indicate validity of dietary monitoring among athletes. The relationships between personality traits and nutritional behaviours of team athletes are not fully unambiguous and require further research.

## Figures and Tables

**Table 1 nutrients-15-00039-t001:** Implementing recommendations of the Swiss pyramid among athletes training in team sports (% of responses).

Nutritional Behaviours	1	2	3	4	5	No 1 + 2	Yes 4 + 5
Consuming at least 3 meals a day	0.0	0.0	2.8	10.3	86.9	0.0	97.2
Regular consumption of meals (every 3–5 h)	0.9	15.5	25.4	46.9	11.3	16.4	58.2
Consuming most caloric meal before/after training	1.4	14.1	29.6	43.7	11.3	15.5	55.0
Consuming 200 mL of vegetable, fruit juice a day	2.3	17.8	36.6	31.5	11.7	20.1	43.2
Consuming vegetables with at least 2 meals a day	6.6	22.1	31.0	31.5	8.9	28.7	40.4
Consuming raw vegetables at least every day	19.7	28.2	36.2	12.7	3.3	47.9	16.0
Consuming 2–3 portions of vegetables every day	8.9	17.8	49.8	16.4	7.0	26.7	23.4
Consuming 1–2 portions of fruit every day	4.2	16.4	46.0	17.8	15.5	20.6	33.2
Cereal products in all main meals	2.3	16.4	56.3	19.2	5.6	18.7	24.8
Wholemeal cereal products min. twice a day	5.2	11.7	50.2	25.8	7.0	16.9	32.8
Milk or dairy products min. twice a day	12.7	27.7	45.1	12.2	2.3	30.4	14.5
Protein products—approx. 150 g/d (2–3 times/week)	12.7	25.8	39.9	12.7	8.9	38.5	21.6
Fish 1–2 times a week	6.6	13.6	43.7	30.0	6.1	20.2	36.1
Limiting animal fats in diet	5.2	13.1	51.6	19.2	10.8	18.3	30.0
Plant-based fats every day (almost every day)	4.7	21.6	50.7	16.0	7.0	26.3	23.0
Adequate hydration during exercise	0.0	0.0	2.8	18.3	78.9	0.0	97.2
Adequate hydration after exercise	0.0	0.0	4.2	30.0	65.7	0.0	95.7
Preferring water for hydration	0.0	3.8	4.7	3.8	87.8	3.8	91.6
Avoiding sweetened carbonated beverages in diet	0.5	5.6	8.5	27.7	57.7	6.1	85.4
Avoiding energy drinks in diet	0.0	2.8	6.6	23.9	66.7	2.8	80.6
Avoiding fast-food products in diet	0.0	3.3	2.8	23.0	70.9	3.3	93.9
Limiting sweet and salty snacks	0.0	22.5	40.4	27.2	9.9	22.5	37.1
Varied diet	15.5	16.0	38.0	21.6	8.9	31.5	31.5

Legend: 1—definitely not, 2—rather not, 3—hard to say, 4—rather yes, 5—definitely yes.

**Table 2 nutrients-15-00039-t002:** Implementing recommendations of the Swiss pyramid and the overall index of proper nutritional behaviours among athletes training in team sports (descriptive statistics).

Nutritional Behaviours	Descriptive Statistics						
	M	SD	Min	Max	Me	Q25	Q75
Consuming at least 3 meals a day	4.84	0.44	3.00	5.00	5.00	5.00	5.00
Regular consumption of meals (every 3–5 h)	3.52	0.92	1.00	5.00	4.00	3.00	4.00
Consuming most caloric meal before/after training	3.49	0.92	1.00	5.00	4.00	3.00	4.00
Consuming 200 mL of vegetable, fruit juice a day	3.32	0.98	1.00	5.00	3.00	3.00	4.00
Consuming vegetables with at least 2 meals a day	3.14	1.07	1.00	5.00	3.00	2.00	4.00
Consuming raw vegetables at least every day	2.52	1.05	1.00	5.00	3.00	2.00	3.00
Consuming 2–3 portions of vegetables every day	2.95	0.99	1.00	5.00	3.00	2.00	3.00
Consuming 1–2 portions of fruit every day	3.24	1.04	1.00	5.00	3.00	3.00	4.00
Cereal products in all main meals	3.09	0.82	1.00	5.00	3.00	3.00	3.00
Wholemeal cereal products min. twice a day	3.18	0.91	1.00	5.00	3.00	3.00	4.00
Milk or dairy products min. twice a day	2.64	0.93	1.00	5.00	3.00	2.00	3.00
Protein products approx. 150 g/d (2–3 times/week)	2.79	1.10	1.00	5.00	3.00	2.00	3.00
Fish 1–2 times a week	3.15	0.96	1.00	5.00	3.00	3.00	4.00
Limiting animal fats in diet	3.17	0.97	1.00	5.00	3.00	3.00	4.00
Plant-based fats every day (almost every day)	2.99	0.92	1.00	5.00	3.00	2.00	3.00
Adequate hydration during exercise	4.76	0.49	3.00	5.00	5.00	5.00	5.00
Adequate hydration after exercise	4.62	0.57	3.00	5.00	5.00	4.00	5.00
Preferring water for hydration	4.76	0.71	2.00	5.00	5.00	5.00	5.00
Avoiding sweetened carbonated beverages in diet	4.37	0.89	1.00	5.00	5.00	4.00	5.00
Avoiding energy drinks in diet	4.54	0.74	2.00	5.00	5.00	4.00	5.00
Avoiding fast-food products in diet	4.62	0.70	2.00	5.00	5.00	4.00	5.00
Limiting sweet and salty snacks	3.24	0.91	2.00	5.00	3.00	3.00	4.00
Varied diet	2.92	1.16	1.00	5.00	3.00	2.00	4.00
Overall index of rational nutrition-based behaviours	81.9	4.64	68.00	91.00	82.00	79.00	85.00
(total)							

Legend: M—arithmetic mean, SD—standard deviation, Me—median, Q25—lower quartile, Q75—upper quartile.

**Table 3 nutrients-15-00039-t003:** Level of personality traits regarding the Five-Factor model among athletes training in team sports (descriptive statistics).

Personality Traits	M	SD	Min	Max	Q25	Me	Q75
Neuroticism	72.15	20.55	23.00	128.0	56.00	71.00	89.00
Extraversion	121.80	15.52	70.00	151.0	111.0	124.0	133.0
Openness	115.00	13.91	92.00	141.0	101.0	115.0	129.0
Agreeableness	123.20	13.14	86.00	146.0	118.0	126.0	132.0
Conscientiousness	128.50	22.22	83.00	168.0	111.0	133.0	144.0

Legend: M—arithmetic mean, SD—standard deviation, Me—median, Q25—lower quartile, Q75—upper quartile.

**Table 4 nutrients-15-00039-t004:** Relationships between personality traits and the overall index of proper nutritional behaviours and the implementation of individual recommendations of the Swiss pyramid among athletes training in team sports (analysis of Pearson’s and Spearman’s R correlation coefficients).

Nutritional Behaviours	N	E	O	A	C
	Pearson’s r				
Overall index of proper nutritional behaviours	0.132 *	−0.027	−0.143 *	0.015	−0.123
	**Spearman’s R**				
Consuming at least 3 meals a day	−0.035	−0.024	−0.052	−0.045	−0.033
Regular consumption of meals (every 3–5 h)	0.143 *	−0.092	−0.092	0.090	0.186 *
Consuming most caloric meal before/after training	0.117	−0.017	−0.119	−0.040	−0.020
Consuming 200 mL of vegetable, fruit juice a day	−0.013	−0.118	−0.056	0.017	−0.037
Consuming vegetables with at least 2 meals a day	−0.053	0.154 *	0.100	−0.126	−0.054
Consuming raw vegetables at least every day	−0.133	0.024	−0.153 *	−0.063	−0.037
Consuming 2–3 portions of vegetables every day	0.147 *	−0.132	−0.132	0.070	−0.049
Consuming 1–2 portions of fruit every day	0.123	−0.004	−0.157 *	−0.036	0.042
Cereal products in all main meals	−0.040	−0.063	−0.050	0.149 *	−0.020
Wholemeal cereal products min. twice a day	−0.015	0.078	0.131	0.039	−0.052
Milk or dairy products min. twice a day	−0.117	−0.039	−0.039	−0.066	0.143 *
Protein products approx. 150 g/d (2–3 times/week)	0.078	0.040	0.035	0.024	−0.056
Fish 1–2 times a week	0.036	−0.008	0.041	−0.003	−0.045
Limiting animal fats in diet	0.042	−0.073	0.001	0.013	0.014
Plant–based fats every day (almost every day)	0.069	0.116	−0.006	0.029	0.086
Adequate hydration during exercise	−0.073	0.093	−0.018	−0.108	0.054
Adequate hydration after exercise	−0.009	0.037	0.012	−0.148 *	−0.022
Preferring water for hydration	0.092	−0.083	0.104	0.034	−0.077
Avoiding sweetened carbonated beverages in diet	−0.073	0.056	−0.089	0.013	0.016
Avoiding energy drinks in diet	0.173 *	−0.004	0.022	0.042	−0.105
Avoiding fast-food products in diet	0.094	0.057	−0.063	0.028	−0.032
Limiting sweet and salty snacks	0.109	−0.079	−0.152 *	0.112	0.148 *
Varied diet	−0.024	−0.035	0.102	0.019	0.001

** p* < 0.05; N—neuroticism, E—extraversion, O—openness, A—agreeableness, C—conscientiousness.

**Table 5 nutrients-15-00039-t005:** Relationships between the personality traits of the Big Five model and the overall index of rational nutritional behaviours among athletes training in team sports (multiple regression analysis).

	R^2^	S_y_	b *	Std. Error b *	B	Std. Error b
Agreeableness	0.994	6.610	0.467	0.037	0.309	0.025
Extraversion			0.355	0.035	0.237	0.024
Neuroticism			0.073	0.022	0.080	0.024
Conscientiousness			0.110	0.033	0.069	0.021

R^2^—multidimensional determination coefficient, S_y_—standard error estimate, b *—standardised partial regression coefficient.

## Data Availability

Not applicable.

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
