# Peer review of "Personality Determinants of Eating Behaviours among an Elite Group of Polish Athletes Training in Team Sports"

_nutrients, 2022, doi:10.3390/nu15010039_

Round 1

Reviewer 1 Report

The manuscript is an interesting original research investigating the personality determinants of eating behavior among Polish athletes. This work contributes to the field of study confirming the Big Five model and allowing the personalization of the dietary education of athletes. Despite that, the manuscript needs some improvements.

The abstract should include a "background" section prior to the aim.

The "conclusions" heading must be removed.

Personally, the keywords should appear in alphabetic order.

The authors should correct some misspellings, such as Line 73, and present all R-values in the same format (with or without space).

Consider including a brief description of the results at the beginning of the discussion.

Author Response

Dear Reviewer,

please see the attachment, 

best regards, MG 

Reviewer 2 Report

Dear Authors,

I believe the work poses an interesting point of view on athlete’s health promotion.

However, some conclusions are not supported by the results and some further methodological details are needed.

 Majors

Line 92: the exclusion criteria should be indicated, in particular for what allergies are concerned.

Line 93: participants are described at the level of the highest league in Poland. To be better understood at international level, I suggest you define the athletic level of your participants according to the following reference: https://doi.org/10.1123/ijspp.2021-0451

Lines 188-189: “Among the personality dimensions of the Five-Factor model, the tested athletes at an advanced sports level obtained the highest results” it doesn’t seem to me that your results have been analysed according to different sport levels. So how can you state the those at an advanced sports level obtained the highest results? This point should be clarified.

Lines 245-247: Your participants did not present health problems despite a non-accurate dietary regimen; therefore, your suggestion is not supported. Please explain further or avoid the assumption that “The noted nutritional mistakes could reduce nutritional and health value of the diet, and indirectly, the health condition and exercise capacity of the athletes.”

Lines 357-358: “the obtained results can be used in planning strategies aimed at changing eating behaviours according to personality traits.” This assumption is not supported by your results since no strategies to change eating behaviours has been tested in the present work. Please avoid unsupported suggestions.

Lines 397-398: “The obtained results indicate validity of dietary monitoring and nutritional education in order to increase rational nutritional choices among athletes.” Again, the assumption is not supported by your results since you covered the dietary monitoring but then did not applied strategies to increase rational nutritional choices. Please avoid unsupported suggestions.

Minor

Line 326: “PE students”, please define the acronym PE

Author Response

Dear Reviewer, 

please see the attachment,

kind regards, MG 

Reviewer 3 Report

Good writing . Very good construction and design

Author Response

Dear Reviewer,

with sincere gratitude and kind regards, MG

Reviewer 4 Report

Dear Authors,
I highly appreciate your work.
It is substantively, methodically and pragmatically correct.
It is a significant contribution to the science of physical culture.
Unfortunately, there were no references to the important works of M. Superson in the discussion.
Therefore, please expand the discussion and complete the references.

Author Response

(The authors gave the same response as above.)

Round 2

Reviewer 2 Report

Dear Authors,

I congratulate you on responding to all my comments, the manuscript can be accepted in the present form.